# Multi-trophic occupancy modeling connects temporal dynamics of woodpeckers and beetle sign following fire

**Morgan W. Tingley**[1]*, **Graham A. Montgomery**[1], **Robert L. Wilkerson**[2], **Daniel R. Cluck**[3], **Sarah C. Sawyer**[4], **Rodney B. Siegel**[2]

**1** Ecology and Evolutionary Biology, University of California, Los Angeles, CA, United States of America,
**2** The Institute for Bird Populations, Petaluma, CA, United States of America, **3** USDA Forest Service, Forest Health Protection, Susanville, CA, United States of America, **4** USDA Forest Service, Vallejo, CA, United States of America

* mtingley@ucla.edu

**Data Availability Statement:** Input data and model code in the R language are available from the Dryad Digital Repository (https://doi.org/10.5068/D1T68M).

## Abstract

In conifer forests of western North America, wildlife populations can change rapidly in the decade following wildfire as trees die and animals respond to concomitant resource pulses that occur across multiple trophic levels. In particular, black-backed woodpeckers (*Picoides arcticus*) show predictable temporal increases then declines following fire; this trajectory is widely believed to be a response to the woodpeckers' main prey, woodboring beetle larvae of the families Buprestidae and Cerambycidae, but we lack understanding of how abundances of these predators and prey may be associated in time or space. Here, we pair woodpecker surveys over 10 years with surveys of woodboring beetle sign and activity, collected at 128 survey plots across 22 recent fires, to ask whether accumulated beetle sign indicates current or past black-backed woodpecker occurrence, and whether that relationship is mediated by the number of years since fire. We test this relationship using an integrative multi-trophic occupancy model. Our results demonstrate that woodboring beetle sign is a positive indicator of woodpecker presence 1–3 years following fire, an uninformative indicator from 4–6 years after fire, and a negative indicator beginning 7 years following fire. Woodboring beetle activity, itself, is temporally variable and dependent on tree species composition, with beetle sign generally accumulating over time, particularly in stands with diverse tree communities, but decreasing over time in *Pinus*-dominated stands where faster bark decay rates lead to brief pulses of beetle activity followed by rapid degradation of tree substrate and accumulated beetle sign. Altogether, the strong connections of woodpecker occurrence to beetle activity support prior hypotheses of how multi-trophic interactions govern rapid temporal dynamics of primary and secondary consumers in burned forests. While our results indicate that beetle sign is, at best, a rapidly shifting and potentially misleading measure of woodpecker occurrence, the better we understand the interacting mechanisms underlying temporally dynamic systems, the more successfully we will be able to predict the outcomes of management actions.

**Funding:** Funding for analysis and research came from the USDA Forest Service. The funders had no role in study design, data collection and analysis, decision to publish, or preparation of the manuscript.

**Competing interests:** The authors have declared that no competing interests exist.

## Introduction

Dead and dying trees are the key resource driving post-fire biodiversity pulses in the western United States, as fire unlocks hundreds of years of stored energy in the form of cellulose and other plant tissue [1]. Fire-killed trees attract insects–particularly woodboring beetles of the families Cerambycidae and Buprestidae–which lay their eggs on or in the bark of dead trees. For many years after a fire, woodboring beetle larvae feed on the accessible wood tissue, forming "galleries" underneath the bark [2]. Beetle larvae are in turn a desirable food resource, and their presence attracts woodpeckers of multiple species, which nest and forage in high densities in recent post-fire forests [3, 4]. The nest holes of woodpeckers subsequently attract and shelter many other wildlife species as well, including owls, flying squirrels, and a variety of secondary cavity-nesting passerine birds [5].

Food resource availability can predict abundance or occurrence of predators, but only if food resource availability is a primary limiting factor of population size [6, 7]. Different bird populations show evidence both for [8–10] and against [11] population limitation due to specific resource availability, and knowledge of the factors limiting species presence can be important for conservation and management. This need for information on limiting factors is especially true for species of conservation concern like the black-backed woodpecker (*Picoides arcticus*), a species which is closely associated with post-fire conifer forests in western North America. Black-backed woodpeckers are known to forage primarily on the larvae of woodboring beetles [4], although recent dietary analysis indicates a wider dietary breadth that commonly includes insects of the orders Diptera, Araneae, and Hymenoptera [12]. Nevertheless, activity and abundance of woodboring beetles declines with time since fire [13], mirroring a temporal pattern of occupancy and abundance that has also been well-documented in black-backed woodpeckers [14, 15], and suggesting that woodboring beetle larvae abundance may fundamentally be driving the post-fire dynamics of black-backed woodpeckers.

A deeper understanding of the spatial and temporal associations in patterns of both woodboring beetles and black-backed woodpeckers could further clarify the degree of linkage between black-backed woodpecker population size and beetle prey abundance. Links between black-backed woodpeckers and their beetle prey would also suggest that monitoring for one may provide inference on the abundance of the other, or that monitoring for one might provide similar inferences about ecological conditions. If a strong link exists, it could enable more efficient assessment of cross-trophic, post-fire biodiversity and ecological integrity from relatively simple, standard surveys. Beetle surveys could thus potentially be used as a simple proxy for woodpecker abundance and/or the ecological conditions woodpeckers may facilitate, although they would not account for other factors, like predation and competition, that might also govern woodpecker population sizes.

Post-fire temporal dynamics of both woodpeckers and woodboring beetles could also depend on forest composition. Ray et al. [13] documented that beetle activity varied by tree species, with trees of the genus *Pinus* decaying more quickly and thus harboring larger numbers of woodboring beetles in the first few years following fire. Black-backed woodpeckers in California have previously been associated primarily with trees of the genus *Pinus* within mixed conifer forests [14], but a mechanistic relationship tying woodpeckers to particular tree species as a function of time since fire has never been clarified.

In 2018, we surveyed for both larval woodboring beetles and black-backed woodpeckers at 128 points arrayed across 22 fires that burned sometime in the previous ten years. Using beetle larvae sign to create a cumulative index of 'food availability', we ask the following two questions: 1) do food resources predict black-backed woodpecker occurrence? and 2) do different tree species differentially provide foraging resources for black-backed woodpeckers in burned

forests? We provide a novel multi-trophic occupancy modeling framework to answer these questions and connect tree species, beetle larvae surveys, and woodpecker detections across burned forests throughout the decade after fire.

## Methods

### Study area and survey methods

We conducted black-backed woodpeckers surveys as part of a long-term project to monitor bird occupancy and trends following forest fire in montane forests of eastern California. Our study area comprised ten contiguous National Forest units within the Sierra Nevada and Southern Cascades ecoregions of California (Fig 1), with forest types dominated by Sierra mixed conifer (primarily *Pinus ponderosa*, *P. lambertiana*, *Abies concolor*, *Pseudotsuga menziesii*, *Calocedrus decurrens*, and *Quercus kelloggii*) and eastside pine (*Pinus ponderosa* dominated with some *P. jeffreyi*). Higher elevation areas contain larger proportions of firs (*A. concolor* and *A. magnifica*) and *Pinus contorta*. In this study region, we randomly selected 50 fires to visit in 2018 that met our sampling criteria of having burned within the previous 10 years and containing at least 50 ha of conifer forest that burned at mid- or high-severity in one of the ten target National Forest units. Many of these fires had been surveyed for black-backed woodpeckers one or more times during the preceding decade, yielding data which we also used in this study. At each of the 50 fires, we randomly generated a target starting location within the fire perimeter, and crews established black-backed woodpecker survey transects beginning as close as possible to the target location. Each transect consisted of approximately 20 survey points, located a minimum of 250 m from each other to minimize double-counting of individuals. All surveys included a 6-minute broadcast survey (subdivided into three 2-minute detection intervals), during which electronic broadcasts of black-backed woodpecker vocalizations and territorial drumming (obtained from The Macaulay Library of Natural Sounds, Cornell Laboratory of Ornithology; recorded by G.A. Keller) were played for 30 seconds, followed by a 1.5-minute silent observation period. At alternating points, broadcast surveys were preceded by a 7-minute passive survey (subdivided into detection intervals of 3, 2, and 2 minutes, respectively). We followed a removal methodology where call broadcasts were suspended after the first detection. We conducted surveys in the morning hours (05:30h–09:30h) between 4 May and 18 July each year.

In 2018, we also conducted targeted woodboring beetle surveys at a subset of 22 of the 50 fires visited for black-backed woodpecker surveys. We used preliminary results from our 2018 black-backed woodpecker surveys to select equal numbers of fires to survey for woodboring beetles in each of 3 categories: fires where we detected black-backed woodpeckers at $> 50\%$ of survey points; fires where we detected black-backed woodpeckers at $< 50\%$ of points; and fires where we detected no black-backed woodpeckers. At each fire we randomly selected six woodpecker survey points at which to conduct woodboring beetle surveys. Up to three of those points were drawn from points where Black-backed Woodpeckers had been detected earlier that summer; the remaining points were drawn from the points at which black-backed woodpeckers were not detected that year.

The six survey points per transect comprise the basic analytical unit for black-backed woodpecker occurrence in our analysis. The mean minimum distance between these points is 472 m. Point count standards generally recommend a minimum inter-point distance of 250 m in order to minimize double-counting, although we acknowledge that the use of broadcast surveys has the potential to attract birds from surrounding areas. However, telemetry work on black-backed woodpeckers in this system shows that independent individuals travel at a velocity of approximately 0.75 km per hour, and a max rate of <1.2 km per hour [16]. With surveys

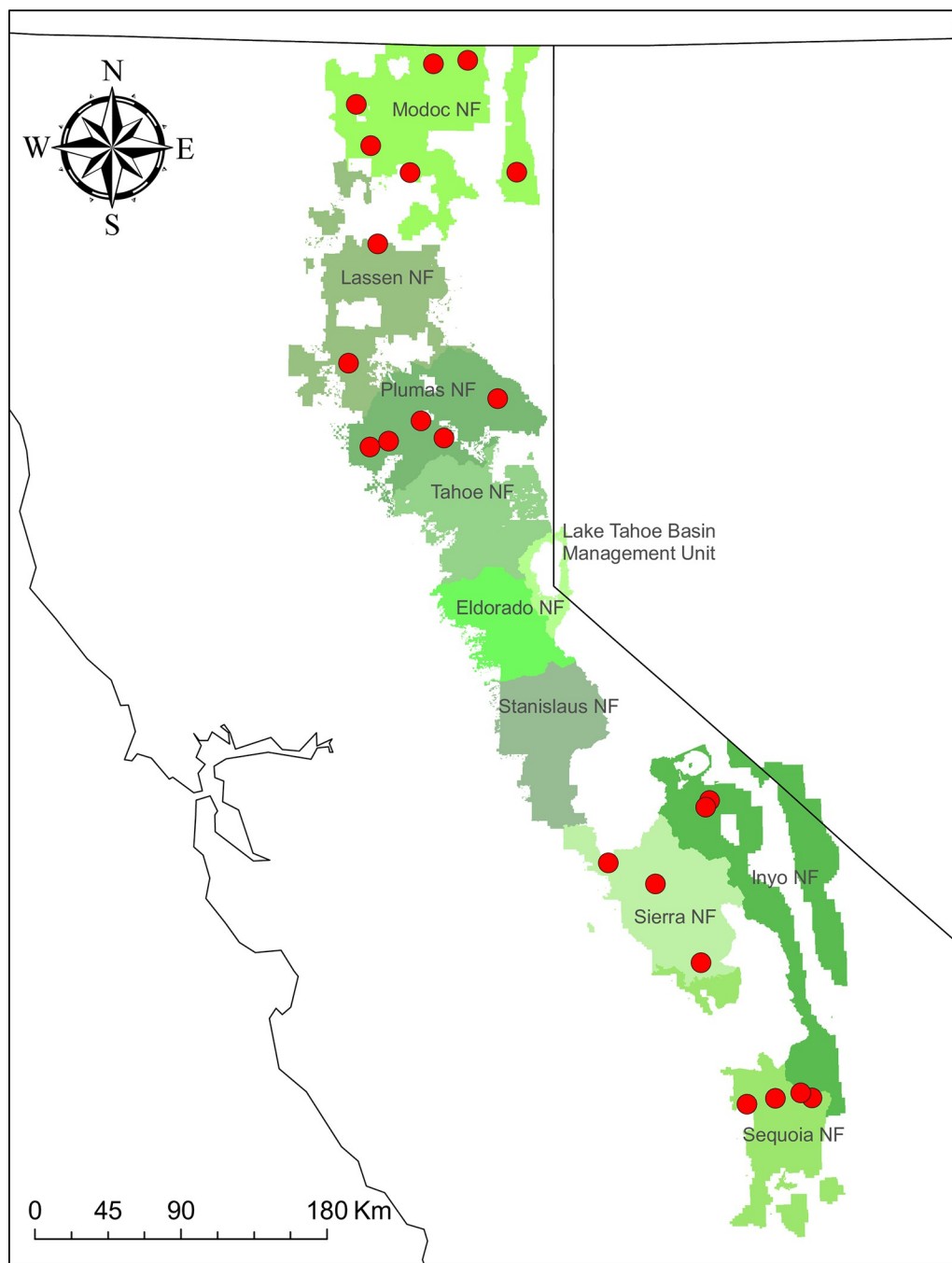

**Fig 1. Locations of the 22 fires (red dots) surveyed for woodboring beetles and black-backed Woodpeckers within U.S. National Forest units in California.** State and forest service boundary polygons courtesy of the U.S. Geological Survey and U.S. Forest Service and reside in the Public Domain.

lasting a median of 7 minutes per point and points separated by approximately 500 m, birds would have to travel approximately twice their maximum normal velocity to keep up with survey broadcasts. Regardless, our analysis of woodpecker occurrence can equally be considered an analysis of "use" should there be any risk of non-independence across survey points [14].

**Table 1. Fires surveyed for both black-backed woodpeckers and woodboring beetles.** Beetle surveys were conducted in 2018 and woodpecker surveys were conducted in 2018 and up to 8 additional years prior.

| Fire Name | Year Burned | U.S. National Forest Unit | No. of Beetle Survey Points | Black-backed Woodpecker Survey Years |
|---|---|---|---|---|
| Aspen | 2013 | Sierra | 6 | 2014, 2015, 2016, 2017, 2018 |
| Bald | 2014 | Lassen | 6 | 2015, 2016, 2017, 2018 |
| Barry Point | 2012 | Modoc | 6 | 2013, 2014, 2015, 2018 |
| Clark | 2016 | Inyo | 6 | 2017, 2018 |
| Cold | 2008 | Plumas | 6 | 2012, 2013, 2014, 2016, 2017, 2018 |
| Cougar | 2011 | Modoc | 6 | 2012, 2014, 2015, 2018 |
| Cove | 2017 | Modoc | 6 | 2018 |
| Fox | 2008 | Plumas | 6 | 2011, 2013, 2014, 2015, 2016, 2017, 2018 |
| Frog | 2015 | Modoc | 6 | 2016, 2017, 2018 |
| George | 2012 | Sequoia | 5 | 2013, 2014, 2015, 2016, 2017, 2018 |
| Granite | 2009 | Sequoia | 6 | 2010, 2011, 2013, 2014, 2015, 2016, 2017, 2018 |
| Lion | 2009 | Sequoia | 6 | 2010, 2011, 2014, 2013, 2015, 2016, 2017, 2018 |
| Minerva 5 | 2017 | Plumas | 6 | 2018 |
| Onion 2 | 2008 | Lassen | 5 | 2010, 2011, 2012, 2013, 2014, 2015, 2016, 2017, 2018 |
| Owens River | 2016 | Inyo | 6 | 2017, 2018 |
| Peak | 2012 | Plumas | 6 | 2015, 2016, 2017, 2018 |
| Pier | 2017 | Sequoia | 6 | 2018 |
| Railroad | 2017 | Sierra | 6 | 2018 |
| Rough | 2015 | Sierra | 4 | 2017, 2018 |
| Scotch | 2008 | Plumas | 6 | 2009, 2010, 2012, 2013, 2014, 2015, 2018 |
| Soup 2 | 2016 | Modoc | 6 | 2017, 2018 |
| Steele | 2017 | Modoc | 6 | 2018 |

Woodboring beetle surveys involved assessing the six closest snags to each selected black-backed woodpecker survey point for larvae activity and sign, and overall condition. Larvae activity and sign were assessed on and under each of two 15 cm x 15 cm bark samples that were removed from the tree, one from the north side of the trunk and one from the south side of the trunk, taken at the DBH line. Each bark sample was given a single, holistic woodboring beetle activity score determined by presence/absence of activity by sample quadrant on bark exterior, interior and sapwood (frass/boring dust, exit/entrance holes, galleries, and presence of larvae). Tree species and physical characteristics (DBH, tree height, high/low bark char heights, needle presence and color) were collected at each snag. Detailed data collection procedures are provided in Ray et al. [13].

Our analysis dataset consisted of surveys for black-backed woodpeckers and beetle activity at 128 points located within 22 fires (Table 1). While beetle surveys were conducted only in 2018, fires ranged in their number of years since fire (1–10) and in the number of years they had been surveyed previously for black-backed woodpeckers prior to 2018 (28 points surveyed in 0 prior years, 24 points in 1 year, 6 points in 2 years, 24 points in 3 years, 6 points in 4 years, 12 points in 5 years, 11 points in 6 years, 12 points in 7 years, and 5 points in 8 years). We used woodpecker survey data from all previous visits– in addition to our 2018 woodpecker and beetle surveys–to model changing woodpecker occurrence over time as a function of beetle sign evident in 2018.

## Analytical approach

We developed a hierarchical model in a Bayesian context to jointly model both the dynamics of beetle activity intensity over time within our plots, as well as the occurrence–accounting for

imperfect detection–of black-backed woodpeckers at those same plots. The model largely follows the structure of a single-species occupancy model [17], where woodpecker observations of detection or non-detection, $y_{jkt}$, for survey interval $k$ at site $j$ (where sites are individual survey points) in year $t$, are assumed to be imperfectly observed representations of the true occurrence status, $z_{jt}$ (present or absent), which is constant across all $k$ survey intervals (i.e., closure is assumed within the <17-minute survey period) but can change from year to year. Observed occurrence of black-backed woodpeckers, $y_{jkt}$, is thus modeled as

$$y_{jkt} \sim \text{Bernoulli}(z_{jt}p_{jkt}), \tag{1}$$

where $p_{jkt}$ is the probability of detection for a given survey at a site. Similarly, the true occurrence status of a site in year $t$, $z_{jt}$, is modeled as

$$z_{jt} \sim \text{Bernoulli}(\psi_{jt}), \tag{2}$$

where $\psi_{jt}$ is the probability of occurrence at a site. In this context, occupancy is defined by the space and time in which the survey is conducted and across which closure is assumed [18]. Thus, our calculated probabilities represent the probability of at least one black-backed woodpecker occurring within (or alternatively, 'using' [19]) the detection radius of a survey point (approximately 120 m) during a survey period (median = 7 minutes in duration).

The probabilities of woodpecker detection and occurrence are both modeled as logit-linear functions of covariates chosen *a priori*. Following previous work studying black-backed woodpeckers with this survey methodology [14, 15, 20], we expected detection, $p_{jkt}$, to vary as a function of an intercept and the linear additive combination of a categorical covariate representing the survey type (passive = 0, broadcast = 1), giving

$$\text{logit}(p_{jkt}) = \alpha_0 + \alpha_{type}\text{type}_k. \tag{3}$$

The probability of woodpecker occupancy of a survey point was modeled as a function of five covariates: (1) elevation, (2) latitude, (3) snag density, (4) intensity of beetle larvae activity (as indirectly measured by cumulative beetle sign since the fire; modeled as a latent variable, see below), and (5) an interaction between years-since-fire and the intensity of beetle larvae activity (with the hypothesis that cumulative beetle sign becomes less predictive over time). Snag counts were conducted immediately after completing woodpecker surveys and consisted of counting all snags of different size classes (10–30, 30–60, and >60 cm dbh) within 50 m of each survey point. Size-specific snag counts were aggregated in the field into different categories ($\leq$5, 6–15, 16–30, 31–50, 51–100, >100), which were converted to numerical quantities (1, 6, 16, 31, 51, 101, respectively) for analysis [15]. Counts across all three size classes were summed to calculate a relative index of snag density (snags/ha). The linear additive model for occupancy in the first year of surveys can be described as,

$$\text{logit}(\psi_{j,t=1}) = \beta_{0,j} + \beta_{elev}\text{elev}_j + \beta_{lat}\text{lat}_j + \beta_{snag}\text{snag}_{jt} + \beta_{beetle}\text{intensity}_{jt}$$
$$+ \beta_{ageXbeetle}\text{age}_{jt}\text{intensity}_{jt}, \tag{4}$$

where $\beta$ represents intercept and slope parameters. To account for pseudoreplication and temporal autocorrelation derived by sampling sites repeatedly in consecutive years, we added a temporal autocorrelation term [21], $\phi$, which was multiplied by the true occurrence status in

year $t-1$, resulting in the following model for additional post-fire years,

$$\text{logit}(\psi_{j,t>1}) = \beta_{0,j} + \beta_{elev}\text{elev}_j + \beta_{lat}\text{lat}_j + \beta_{snag}\text{snag}_{jt} + \beta_{beetle}\text{intensity}_{jt} + \beta_{ageXbeetle}\text{age}_{jt}\text{intensity}_{jt}$$
$$+ \phi z_{j,t-1} \tag{5}$$

As 2018 was the last year of surveys used in this dataset, and also the only year with *in situ* beetle activity surveys, all surveys conducted in 2018 held the temporal index of $t = 10$. Surveys in previous years ($t = 1,\ldots,9$) were treated as missing data if no surveys occurred at a site in that survey-year. Finally, the intercept ($\beta_{0,j}$) was modeled as a random effect for each fire (n = 22), drawn from a hierarchical normal informed by a common mean ($\mu_{\beta0}$) and precision ($\tau_{\beta0}$).

Previous analyses of black-backed woodpecker occurrence have demonstrated the importance of the number of years since fire in models of the species' occurrence [14, 17], yet our model of woodpecker occupancy does not include an independent effect of time since fire (Eqs 4–5). By excluding time since fire from our model of woodpecker occupancy, we assume that all temporal changes in occupancy are due to either (1) temporal changes in habitat quality as indicated by intensity of beetle sign (parameters $\beta_{beetle}$ and $\beta_{ageXbeetle}$), or (2) random stochasticity (the frequency of which is governed by the parameter $\phi$).

A novel feature of our multi-trophic model is that we treat cumulative beetle larvae sign at a survey point each year (intensity$_{jt}$) as a latent (i.e., indirectly observed) variable, which for mathematical simplicity we define as continuous. We are then able to model beetle larvae sign as a function of different environmental variables expected to relate to beetle activity and to account for the known dynamic that beetle sign generally accumulates over time even though overall activity may decline. Thus, we hypothesized that the intensity of beetle sign at a site each year (intensity$_{jt}$) varies as a function of: (1) the number of years since fire; (2) the proportion of sampled trees per point that were of the genus *Pinus*; (3) and an interaction between the proportion of pines and years since fire. Based on previous work [13], we hypothesized that beetle sign increases over time (as sign is generally cumulative and lasting), but that pines would have greater activity in early post-fire years and lower activity in later post-fire years (as pine bark generally decomposes faster than bark of other trees in our study areas). We thus modeled beetle sign intensity as,

$$\text{logit}(intensity_{jt}) = \gamma_0 + \gamma_{age}\text{age}_{jt} + \gamma_{pine}\text{pine}_j + \gamma_{ageXpine}\text{age}_{jt}\text{pine}_j. \tag{6}$$

We fit this model to observed data collected in 2018, by treating the total sum of beetle sign scores across all surveyed trees per point (max = 6) as a binomially distributed variable, as follows,

$$\text{activity}_j \sim binomial(intensity_{j,t=10}, \text{numTrees}_j * 8), \tag{7}$$

where the maximum activity score is a product of the number of trees sampled per point (numTrees$_j$) and the maximum potential activity score per tree (i.e., 8).

We fit the model to the data with JAGS [22] using the R statistical programming language version 4.0.2 [23] and the package 'R2jags' [24]. We used vague priors (i.e., normal with $\mu = 0$, $\tau = 0.1$). We ran three chains of 50,000 iterations thinned by 50 with a burn-in of 50,000, yielding a posterior sample of 3,000 across all chains. Convergence was checked visually with traceplots and confirmed with a Gelman-Rubin statistic $< 1.1$ [25]. Inference on parameters was made using 95% Bayesian credible intervals (95 CI).

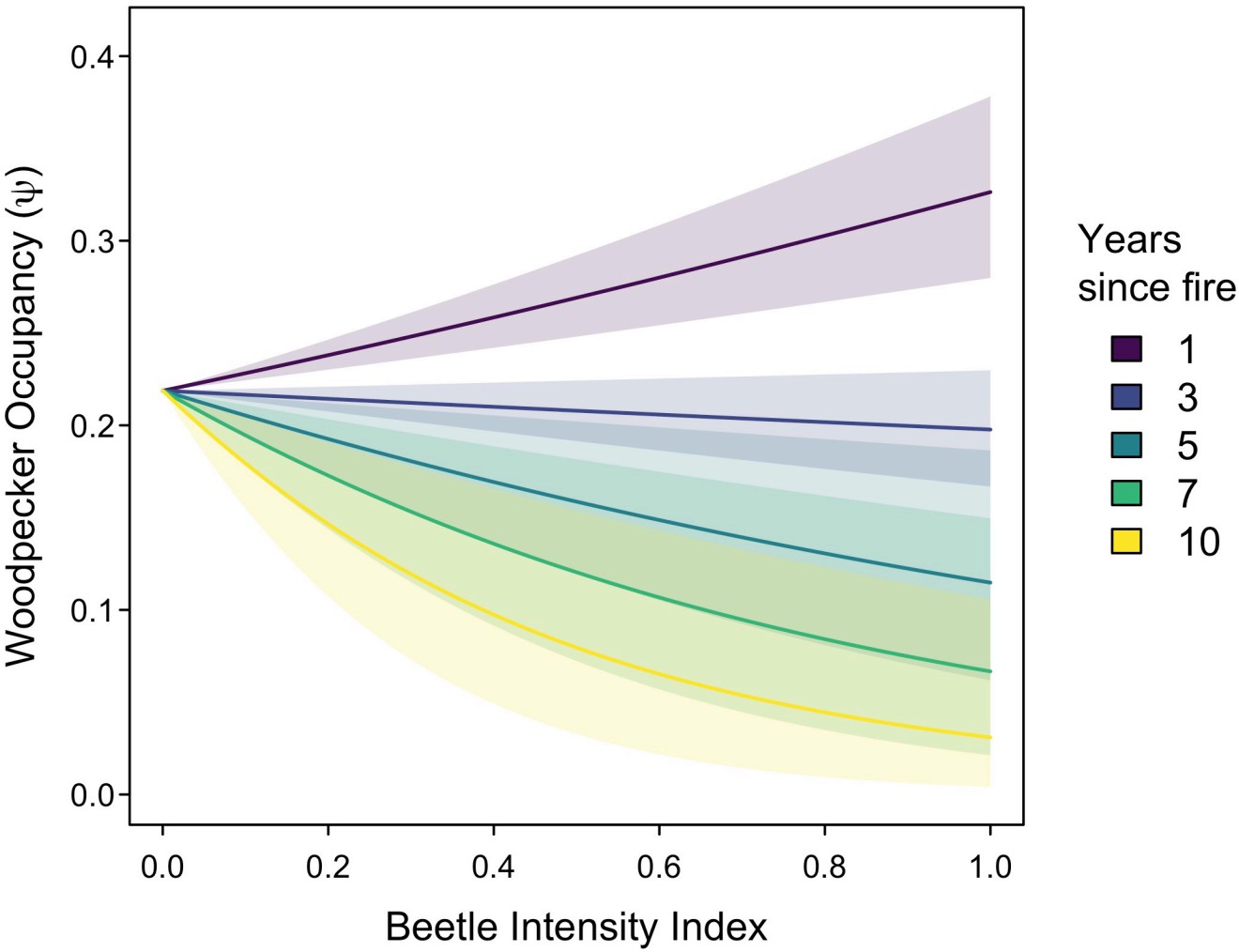

**Fig 2. Black-backed woodpecker occupancy as a function of an index of intensity of woodboring beetle sign.** The relationship differs depending on the number of years since fire (only 5 years shown, for clarity). Solid lines represent posterior means and ribbons represent partial 95% credible intervals representing uncertainty only in the interaction between beetle sign and years since fire.

## Results

### Do food resources predict recent black-backed woodpecker occurrence?

We found no temporally consistent relationship between woodpecker occupancy and the intensity of beetle sign ($\beta_{beetle}$ = 0.19, 95% CI = -3.09, 3.70), but a strong interactive relationship of beetle sign intensity with fire age on woodpecker occupancy ($\beta_{ageXbeetle}$ = -0.95, 95% CI = -1.56, -0.37). This strong relationship indicates that in the first few years following fire, beetle sign is a positive indicator of black-backed woodpecker occupancy, but by 10 years after fire, beetle sign is a negative indicator of black-backed woodpecker occupancy (Fig 2).

### Do different tree species differentially provide foraging resources for black-backed woodpeckers over time in post-fire forests?

Consistent with our hypotheses, beetle sign was higher in plots with a greater proportion of pine trees ($\gamma_{pine}$ = 0.36, 95% CI = 0.20, 0.53) and generally increased over time following fire ($\gamma_{age}$ = 0.26, 95% CI = 0.20, 0.32). However, we found a strong negative interaction between

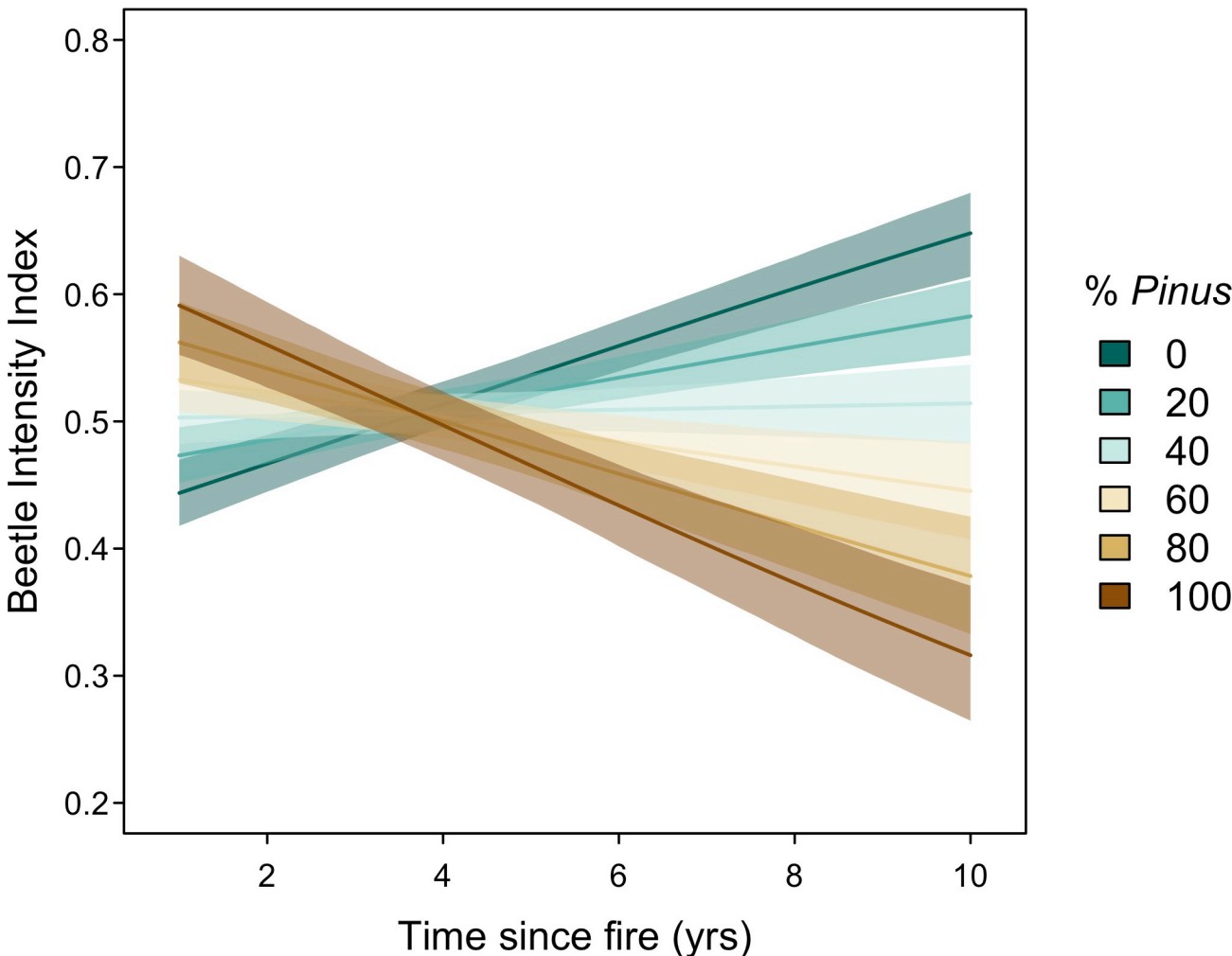

**Fig 3. Index of intensity of woodboring beetle sign as a function of the number of years post-fire and the percentage of trees of the genus *Pinus* sampled in each plot.** Solid lines represent posterior means and ribbons represent 95% credible intervals around the predicted intensity index.

these two variables on beetle sign intensity ($\gamma_{ageXpine}$ = -0.62, 95% CI = -0.75, -0.49), such that beetle sign increases over time in plots that are primarily trees other than of the genus *Pinus*, but actually decreases over time when plots are mostly pine trees (Fig 3).

### Additional findings

We found positive associations of occupancy with higher elevation and latitude, weak evidence for a positive association of occupancy with snag density (controlling for beetle activity), and strong evidence for association of survey type (passive vs. broadcast surveys) on detectability (Table 2).

### Discussion

We found strong evidence for relationships between woodpecker occurrence and the intensity of beetle sign; however, the magnitude and direction of these relationships changes rapidly over time. Beetle sign 1–3 years post-fire appears to be a good indicator of occupancy by black-backed woodpeckers during that time period–perhaps even a better indicator than snag

**Table 2. Posterior estimates of model parameters.** Slope parameters with 95% Bayesian credible intervals that do not cross zero suggest relationships for which we have strong evidence and are highlighted in bold.

| Response | Parameter | Estimate | Lower 95% CI | Upper 95% CI |
|---|---|---|---|---|
| Woodpecker detection | Intercept | -5.17 | -7.13 | -3.77 |
| - | **Survey type** | 5.47 | 4.01 | 7.37 |
| Woodpecker occupancy | Global intercept ($\mu_{\beta0}$) | -1.27 | -3.06 | 0.43 |
| - | Beetle sign | 0.19 | -3.09 | 3.70 |
| - | **Beetle sign * fire age** | -0.95 | -1.56 | -0.37 |
| - | **Elevation** | 1.25 | 0.65 | 1.92 |
| - | **Latitude** | 0.93 | 0.27 | 1.47 |
| - | Snag density | 0.14 | -0.03 | 0.30 |
| - | Temporal autocorrelation | 0.68 | -0.09 | 1.47 |
| Beetle activity | Intercept | -0.13 | -0.22 | -0.04 |
| - | **Proportion of pines** | 0.36 | 0.19 | 0.53 |
| - | **Years since fire (fire age)** | 0.26 | 0.20 | 0.32 |
| - | **Proportion of pines * fire age** | -0.62 | -0.75 | -0.49 |

density (which is often used as a spatial indicator of woodpecker occupancy within fires) [14, 20, 26]. However, between 4–7 years post-fire, the signal from beetle sign becomes muddled, with little to no relationship between beetle sign and woodpecker occupancy. By 10-years post-fire, the relationship has reversed, with areas of high beetle sign showing lower black-backed woodpecker occupancy.

The likely reason behind this inverse relationship is that beetle activity sign should monotonically accumulate over time in the first decade following fire (until trees decay so much that sign deteriorates), even if beetle activity, itself, strongly declines with time [27, 28]. For example, a particular tree may only contain large numbers of beetle larvae for the first 3–4 years following fire but may continue to accumulate beetle activity sign gradually for a full decade. Consequently, by 10 years post-fire, the accumulated sign of woodboring beetles may be a better indicator of past–rather than present–black-backed woodpecker occurrence.

The expected pattern of accumulation of beetle sign is compounded by temporal dynamics of beetle activity and wood decay that differ by tree species. In the middle elevations of the Sierra Nevada, where most canopy conifers are either of the genus *Pinus* or *Abies* (with patches of *Pseudotsuga* and *Calocedrus*), woodboring beetle temporal dynamics can differ markedly by tree species [13], as different tree genera decay and fall at different rates, while also subject to variation owing to tree size and local conditions [29–31]. Trees of the genus *Pinus* generally have vascular tissue that quickly becomes suitable for feeding larvae following mortality, leading to rapid post-fire colonization by woodboring beetles [30]. As part of the decay process, pine bark sloughs off relatively quickly, as the vascular tissue dries out or is completely consumed by beetle larvae. The loss of bark leaves a hardened tree bole exposed to the elements and unsuitable for larvae, vastly reducing the role of *Pinus* trees as beetle larvae reservoirs several years after death. Vascular tissue of other trees such as those of the genus *Abies*, by comparison, does not become suitable for woodboring beetles as quickly following death [30] – possibly due to different under-bark microclimates or differences in host volatile attraction [13] –so plots with primarily non-*Pinus* trees show patterns of beetle intensity that increase over time, rather than decrease (Fig 3). The temporal difference in beetle suitability between *Pinus* and *Abies* trees may be further exacerbated by the long-term fate of snags of each. *Pinus* snags tend to fall over after death–possibly due to destabilization by a woodboring beetle that targets pine roots interacting with appropriate decay microclimate and fungi [31] –which

could limit beetle colonization and woodpecker foraging, while *Abies* snags tend to snap halfway or near the crown, providing longer-term foraging resources [32–35].

As a methodological note, we were able to uncover these relationships due to the development of a novel multi-trophic occupancy model. Multi-species occupancy models primarily either treat multiple detected species as occurring independently in a random-effects framework [36, 37] or directly interacting with the occupancy and/or detectability of other species in frameworks meant to model just a handful of species [38, 39]. In both cases, however, models generally assume that all included species are detected or surveyed through the same type of method and at identical spatial and temporal sampling scales. In our system, woodpecker occurrence (measured via point counts) is potentially influenced by woodboring beetle abundance, which is assumed to be unaffected by woodpecker occurrence; but critically, woodboring beetle abundance is indirectly assessed via accumulating beetle sign from up to six trees at each woodpecker survey point. Thus, our model integrates survey data on two trophic levels [40] within a hierarchical system where abundance of the lower trophic level potentially impacts the occurrence of the higher trophic level. Such a model structure could be easily generalized to other multi-trophic systems, where survey assessment of each trophic level generally follows different methodologies each with its own unique observation process.

## Conclusion

Rapid and reliable survey methodologies are critical for land managers tasked with making decisions following fire across large land expanses. While systematic bird surveys (e.g., point counts) are well established monitoring tools, woodboring beetle surveys in post-fire areas have held an intriguing allure due to their potential to provide multi-trophic inference on both insects and their predators (e.g., woodpeckers), as well as the ecological conditions the presence of these species may facilitate. Our beetle survey methods–which integrated assessments of frass, boring dust, exit and entrance holes, galleries, and direct counts of larvae–are comprehensive yet complicated by the cumulative nature of nearly all these forms of beetle sign (larvae counts being the exception). This accumulation of sign interacts with the decomposition and decline of the structural integrity of snags, which ultimately leads to a complex and non-linear relationship of beetle sign with woodpecker occurrence (Fig 2). Ultimately, based on these results, beetle sign in general should not be used as a proxy for woodpecker abundance after 2 or 3 years following fire, and may also not be a good proxy for beetle abundance after that point (although we did not directly assess beetle abundance). In the initial 1–2 years following fire, when rapid management decision-making is often most critical, however, beetle sign may be a reliable method for assessing immediate multi-trophic responses to post-fire conditions and related aspects of ecological integrity. Nevertheless, given the non-linear temporal dynamics of black-backed woodpeckers in post-fire forests [15], combined with the apparent shifting usage of tree species with time since fire both for woodboring beetles and their predators (i.e., from *Pinus* to *Abies*), the portions of burned forests that harbor high beetle and/or black-backed woodpecker abundances immediately after fire may not sufficiently sustain black-backed woodpecker populations over the longer term. Together, these results imply that management activities for black-backed woodpeckers should account not only for where woodpeckers are when post-fire forest management actions are implemented (i.e., usually within 1–3 years following fire), but also consider, as feasible, where the woodpeckers are likely to go in the near future given a shifting mosaic of tree mortality and prey availability.

The temporally-variable but strong relationship between woodpecker occurrence and beetle activity sign ultimately corroborates the strong association between black-backed woodpeckers and woodboring beetles in western forests–a keystone consideration for the management and

conservation of black-backed woodpeckers. While this is perhaps an unsurprising finding given prior work in this system and species [1, 20, 27, 41, 42], its confirmation is not trivial, as many bird species do not show such prey-dependence [7] and black-backed woodpecker diets show a great diversity of insect prey [12]. While many previous studies have shown strong positive relationships between black-backed woodpeckers and availability of dead trees [14, 20, 42–45], we found no association of snag density on woodpecker occupancy while simultaneously accounting for beetle sign (Table 2). Snag density has long been considered a proxy for food availability in this system [45], even though snags are also used predominantly for nesting [46, 47]. Our results thus confirm what has long been indirectly hypothesized about black-backed woodpeckers, that their fine-scale spatial distribution is strongly impacted by the spatiotemporal dynamics of prey availability, which itself varies non-linearly over time following fire and as a function of forest tree composition.

## Acknowledgments

We are grateful to the numerous field technicians who have gathered woodpecker occurrence data over many years as part of the Management Indicator Species program. The manuscript was improved by input from Dárius Tubelis and two anonymous reviewers. This manuscript is Contribution No. 747 of The Institute for Bird Populations.

## Author Contributions

**Conceptualization:** Robert L. Wilkerson, Daniel R. Cluck, Sarah C. Sawyer, Rodney B. Siegel.

**Data curation:** Robert L. Wilkerson.

**Formal analysis:** Morgan W. Tingley, Graham A. Montgomery.

**Funding acquisition:** Sarah C. Sawyer, Rodney B. Siegel.

**Investigation:** Robert L. Wilkerson, Daniel R. Cluck.

**Methodology:** Morgan W. Tingley.

**Project administration:** Rodney B. Siegel.

**Visualization:** Morgan W. Tingley, Robert L. Wilkerson.

**Writing – original draft:** Morgan W. Tingley, Graham A. Montgomery.

**Writing – review & editing:** Robert L. Wilkerson, Daniel R. Cluck, Sarah C. Sawyer, Rodney B. Siegel.

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
