## [Decision Letter · Decision Letter 0]

22 Nov 2022

PONE-D-22-28125Multi-trophic occupancy modeling connects temporal dynamics of woodpeckers and beetle sign following firePLOS ONE

Dear Dr. Morgan Tingley,

Thank you for submitting your study to PLOS ONE.

You have produced a high-quality research paper, and reviewers were constructive (they requested Minor and Major reviews). Please try to follow their suggestions, and present explanations when you disagree.

 Reviewer 1 was quite positive about your submission, and made some comments and suggestions.

 Reviewer 2 is mainly concerned with the independence of the survey stations. It would be good to provide some explanations/details or make changes to avoid negative criticism by readers. This reviewer also made several comments and suggestions on the manuscript text (please see the yellow balloons).

 I also made some corrections regarding formatting and presentation (please see below).

We look forward to receiving your revised manuscript.

Kind regards,

Dárius Pukenis Tubelis, Ph.D.

Academic Editor

PLOS ONE

Journal Requirements:

4. We note that Figure 1 in your submission contain map image which may be copyrighted. All PLOS content is published under the Creative Commons Attribution License (CC BY 4.0), which means that the manuscript, images, and Supporting Information files will be freely available online, and any third party is permitted to access, download, copy, distribute, and use these materials in any way, even commercially, with proper attribution. For these reasons, we cannot publish previously copyrighted maps or satellite images created using proprietary data, such as Google software (Google Maps, Street View, and Earth). For more information, see our copyright guidelines: http://journals.plos.org/plosone/s/licenses-and-copyright.

Review by the Editor Dárius Tubelis

Lines 77, 141. It should be “Ray et al [13]” (italics and no dot) (it is the number, not the year). Please check if it happened with other citations (I did not find).

Lines 97. It should be “(Fig 1)”. (no dot). Please check this for all citations and captions of figures.

Line 276. Maybe, “;” should be replaced by “)”. It is strange.

Reference 7. Initials should not be in capitals, except in the first word, and names of regions, species, etc…Refs 16, 39 and 41 also have this problem. Please check all refs carefully again.

References 20 and 21. The underline should not be there.

Reference 32. It appears to have a problem with the second author.

Reference 33. The journal is missing. Add: “Trends Ecol Evol.”

Reference 40. The species name should be in italics. Please check this for all refs.

Reference 43. Use the abbreviated name of the journal. Please check this too.

Reviewers' comments:

Reviewer's Responses to Questions

**Comments to the Author**

1. Is the manuscript technically sound, and do the data support the conclusions?

Reviewer #1: Yes

Reviewer #2: No

2. Has the statistical analysis been performed appropriately and rigorously? 

Reviewer #1: Yes

Reviewer #2: No

3. Have the authors made all data underlying the findings in their manuscript fully available?

Reviewer #1: Yes

Reviewer #2: Yes

4. Is the manuscript presented in an intelligible fashion and written in standard English?

Reviewer #1: Yes

Reviewer #2: Yes

5. Review Comments to the Author

Reviewer #1: Manuscript "Multi-trophic occupancy modeling connects temporal dynamics of woodpeckers and beetle sign following fire" is a unique set of long-term data that sheds light on the relationship between different components of ecosystems. The manuscript is of particular interest in view of the fact that the data obtained in it can be used both in protection and in management. Statistical analysis in these ms been performed appropriately and rigorously. Manuscript written in good English. This ms certainly deserves publication in this journal. Despite this, I have a number of comments and suggestions.

1. Keywords. The article talks a lot about dead and dying trees, perhaps the authors should include this in the keywords, instead of repeating the name of the woodpecker in English and Latin?

L 117 - Were fluctuations taken into account in the long-term number of woodpeckers? As studies show, the food supply is not the only factor affecting the number and occurrence of woodpeckers. The number of beetles can be very high, and there can be no woodpeckers due to low numbers, for example, due to poor breeding or a harsh winter that led to a strong decrease in numbers.

L 187-189. Sorry, but are you sure this is the best way to convert categorical variables to scale variables?

L 205-207. This is very interesting and novel, but I have some doubts that it is "continuous"

Reviewer #2: One of my main concerns is that survey stations are used as the independent unit of analysis.

Additional information is required to justify this decision.

For example, what was the average distance between survey stations (and standard deviation)? Did this distance vary based on the size of the fire perimeter?

As the manuscript is written, this critical issue does not receive sufficient discussion.

6. PLOS authors have the option to publish the peer review history of their article (what does this mean?). If published, this will include your full peer review and any attached files.

Reviewer #1: No

Reviewer #2: No

---

## [Author Response · Author response to Decision Letter 0]

23 Jan 2023

Please see the uploaded "Response to Reviewers" document which carefully outlines all responses and revisions

---

## [Editor Report · Decision Letter 1]

30 Jan 2023

Multi-trophic occupancy modeling connects temporal dynamics of woodpeckers and beetle sign following fire

PONE-D-22-28125R1

Dear Dr. Morgan Tingley,

We’re pleased to inform you that your manuscript has been judged scientifically suitable for publication and will be formally accepted for publication once it meets all outstanding technical requirements.

Kind regards,

Dárius Pukenis Tubelis, Ph.D.

Academic Editor

PLOS ONE

Additional Editor Comments:

Dear Dr Morgan W. Tingley and co-authors,

I received the corrected version of your manuscript five days ago. I consider that it can be accepted in definitive for publication in PLOS ONE.

This is because you properly made changes in the text, following the reviewer´s suggestions, besides some other minor changes, besides a new analysis. 

I also appreciated the responses that you provided to both reviewers. They are very convincing.

Thus, I think that you produced an excelent paper, reflecting a well done research in this field of forest management.

Please follow the instructions that you will receive from the PLOS ONE team to complete the publication proccess.

Please make the following changes when you receive the proofs for correction, or before it, if possible:

Line 104. Eliminate the italics of "and".

Line 121. Maybe, it would be better like "05:30h"...

About using the verb "hypothesize" (suggested by an reviewer).

Please note that it still persists in other parts of the text.

Line 235. Maybe you can replace by "variables supposed to". Could be ?

Lines 237 and 240. Maybe, you can use something like "our hypothesis was that....". Please do it if you agree.

Acknowledgments (line 402):

Do not forget to add the number of the publication (I refer to XXX) of your Institute.

In the References Section:

Ref 1. There is a DOI, please add "doi:10.1890/08-0895.1"

Ref 4. Use the abbreviated title of the journal.

Ref 11. Delete "Storch, editor".

Ref 14. Delete the space before "backed".

Ref 15. Delete the space before "scale".

Ref 18. Add the city and ": " before Academic Press.

Ref 19, Use the abbreviated name of the journal.

Ref 25. Add the city and ": " before CRC/.

Ref 29. Use the abbreviated title of the journal.

Ref 32. Replace the capitals by small letters.

Ref 36. Use the abbreviated title of the journal.

Ref 38. The same above.

Ref 42 and 43. Delete the traces "-" between the capitals of author´s names.

---

## [Editor Report · Acceptance letter]

27 Feb 2023

PONE-D-22-28125R1 

Multi-trophic occupancy modeling connects temporal dynamics of woodpeckers and beetle sign following fire 

Dear Dr. Tingley:

I'm pleased to inform you that your manuscript has been deemed suitable for publication in PLOS ONE. Congratulations! Your manuscript is now with our production department. 

Kind regards, 

on behalf of

Dr. Dárius Pukenis Tubelis 

Academic Editor

PLOS ONE